# Monitoring Mushroom Growth with Machine Learning

**Vasileios Moysiadis** [1] , **Georgios Kokkonis** [2] , **Stamatia Bibi** [1] , **Ioannis Moscholios** [3] , **Nikolaos Maropoulos** [1] and **Panagiotis Sarigiannidis** [1,*]

1   Department of Electrical and Computer Engineering, University of Western Macedonia, 50100 Kozani, Greece
2   Department of Business Administration, University of Western Macedonia, 51100 Grevena, Greece
3   Department of Informatics and Telecommunications, University of Peloponnese, 22100 Tripolis, Greece
*   Correspondence: psarigiannidis@uowm.gr

**Abstract:** Mushrooms contain valuable nutrients, proteins, minerals, and vitamins, and it is suggested to include them in our diet. Many farmers grow mushrooms in restricted environments with specific atmospheric parameters in greenhouses. In addition, recent technologies of the Internet of things intend to give solutions in the agriculture area. In this paper, we evaluate the effectiveness of machine learning for mushroom growth monitoring for the genus *Pleurotus*. We use YOLOv5 to detect mushrooms' growing stage and indicate those ready to harvest. The results show that it can detect mushrooms in the greenhouse with an F1-score of up to 76.5%. The classification in the final stage of mushroom growth gives an accuracy of up to 70%, which is acceptable considering the complexity of the photos used. In addition, we propose a method for mushroom growth monitoring based on Detectron2. Our method shows that the average growth period of the mushrooms is 5.22 days. Moreover, our method is also adequate to indicate the harvesting day. The evaluation results show that it could improve the time to harvest for 14.04% of the mushrooms.

**Keywords:** mushroom; YOLOv5; Detectron2; machine learning; object detection; instance segmentation

## 1. Introduction

Mushrooms are a rich source of nutrients, proteins, minerals, and vitamins (B, C, and D) [1]. More than 200 species of edible mushrooms are used as ingredients in traditional foods around the world [2], but only 35 species are cultivated in greenhouses in restricted environments [3]. The mushrooms of the genus *Pleurotus* are in second place worldwide in the industry. Therefore, improving the production line will have a major impact on the economy of the specific domain. The authors in [4] presented a comprehensive review of the factors affecting the mushrooms of the genus *Pleurotus* spp. In addition, many research efforts have focused on solutions based on modern technologies from the Internet of things, aiming to transform traditional farming into the new era called smart farming [5]. This transition brings many applications in various agriculture tasks aiming to increase production, reduce cost production, reduce chemical inputs, and reduce labour effort.

Along with other technologies, machine learning is highly used in tasks such as yield prediction, disease detection, weed recognition, and fruit recognition. More specifically, many research efforts try to classify and detect the specific location of different objects in images. For example, a robotic mechanism is suitable for weeding if it has the ability to recognise weeds from vegetation [6]. The detection of fruits with machine learning is also valuable in robotic mechanisms [7–9]. Disease detection has already seen numerous research efforts evaluating the ability to detect various diseases in various crops [10–12]. Moreover, insect detection is another important task in cultivation. Thus, many research efforts exist in this area [13,14].

Furthermore, machine learning is already used in research efforts in mushroom cultivation or wild mushroom hunting, aiming to give solutions to various tasks. The authors in [15] gave a comprehensive review in this area. One of the main tasks based on machine

learning focuses on species recognition. For example, the authors in [16] provided a method to distinguish edible mushrooms from poisonous ones. In addition, the authors in [17] suggested a method for species recognition with a mobile smart device. Another research effort on mushroom species classification is presented in [18]. Furthermore, a few research efforts have used object detection algorithms in order to localise and classify mushrooms in a photo. The authors in [19] developed a system that used object detection to localise and recognise mushrooms that were ready to harvest. In [20], the authors evaluated the potential of mushroom hunting using a custom-made unmanned aerial vehicle (UAV). Finally, the authors in [21] presented an automatic robotic mechanism for mushroom harvesting using machine learning for detection.

Object detection algorithms have gained significant momentum in recent years due to the success of deep learning [22], a specific area of machine learning. As a result, many object detection algorithms have been available in recent years that can detect and provide the exact location of an object in an image or video. In addition, some of them are also capable of providing a corresponding mask of the object. This category is called instance segmentation. Detectron2 [23] is the most popular in this category, while some other variants, such as Mask Scoring RCNN [24], try to return better results of the provided masks. In addition, YOLOv5 [25] is one of the most popular object detection algorithms, delivering accurate and rapid results.

With recent technologies, a sufficient number of research efforts are available for mushroom cultivation. Most of them propose controlling growing conditions in the greenhouse and keeping them within specific boundaries [26,27]. However, only a few of the research efforts go a step over and use more advanced mushroom cultivation methods. Machine learning promise to provide efficient methods in this direction. Thus, we should use it to help farmers in their everyday activities and support their decisions. Until now, only a few research efforts have used machine learning in the mushroom industry. As described before, most of them deal with mushroom species classification, and only a few try to give solutions for usable tasks such as mushroom harvesting in the greenhouse. Moreover, none of the existing methods for mushroom harvesting utilises the genus *Pleurotus*.

Our work uses machine learning not only to detect the mushrooms in the greenhouse but also to classify them in three different stages depending on their growth status and predict when they are ready to harvest. For this purpose, we use the YOLOv5 [25] object detection algorithm and evaluate its accuracy on mushroom detection with different configurations of hyperparameters. The evaluation results show that YOLOv5 can detect and classify mushrooms with an F1-score of up to 76.5% and detect the final stage of mushroom growth with an accuracy of up to 70%.

Furthermore, in the second part of our research, we use Detectron2 [23] to extract a corresponding mask for each mushroom and use it to calculate its size and monitor the growth rate. The results show that they follow a linear growth rate, and it is also possible to predict the harvesting day based on images on previous days. In addition, the method can improve the harvesting time for 14.04% of the mushrooms.

Our work could be a valuable method for mushroom detection on greenhouses with practical applications. For example, it could be used for yield prediction, where ground cameras can observe the greenhouse and predict the yield a few days before the mushrooms are ready to harvest. Another possible application could be on robotics mechanisms that can detect the exact crown of the mushrooms in order to harvest them with no damage. This work is part of our decision support system described in our previous work [28], where we presented a system architecture that covered mushroom cultivation in a greenhouse and the aggregation of useful information for wild mushroom hunting.

The rest of this paper is as follows. In Section 2, we give the most relevant research efforts for mushroom detection using machine learning or image processing. Section 3 briefly discusses the basic concepts of object detection and instance segmentation and provides the essential features of YOLOv5 and Detectron2. Section 4 analyses the methodology we follow in this research. Next, Section 5 presents the evaluation results between

different configurations of YOLOv5 and its effectiveness on mushroom classification at different stages of mushroom growth. In addition, we provide the results from Detectron2 on mushroom growth monitoring. Section 6 discusses the effectiveness of the proposed methods, their limitations and possible impact on future applications. Finally, Section 7 concludes this paper.

## 2. Related Work

In this section, we briefly present similar research efforts in the area of smart farming, particularly in image processing or machine learning for image segmentation or object recognition on mushroom cultivation.

A harvesting robot for oyster mushrooms was proposed in [21]. The development system used an improved SSD algorithm to detect mushrooms ready to harvest. The algorithm used RGB images and point clouds collected by an Intel RealSense D435i camera. The evaluation results showed an accuracy of mushroom recognition of up to 95.0% and a harvesting success rate of up to 86.8%. The average harvesting time for a single mushroom was 8.85 s.

The authors in [19] provided a measurement monitoring system to observe mushroom growth in a greenhouse. The proposed system used YOLOv3 for mushroom detection and an additional localisation method to improve the position of the detected mushroom in order to distinguish the same mushroom in different captured photos. Moreover, the system was able to estimate the harvesting time.

In [16], the authors presented a mushroom farm automation to classify toxic mushrooms. The proposed system adopted machine learning to distinguish edible mushrooms from poisonous mushrooms. The utilised model was a combination of six different classifiers that worked together and reach an accuracy of 100%. More specifically, the ensemble model used the following classifiers: decision tree (DT), logistic regression (LR), K-nearest neighbours (KNN), support vector machine (SVM), naive Bayes (NB), and random forest (RF). Moreover, the manuscript introduced an architectural design for smart mushroom farming.

The authors in [29] presented an automatic sorting system for fresh, white button mushrooms. Apart from the automatic mechanism, they proposed an image processing algorithm to collect button mushrooms. The algorithm eliminated the shadow and petiole on the image and determined the pileus diameter of the mushrooms. Experimental results showed that compared to manual grading speed, their approach was improved by 38.86%, and the accuracy was improved by 6.84%.

## 3. Background

In this section, we briefly presenting the main characteristics of YOLOv5 [25] and Detectron2 [23]. Both of them are suitable for object detection in images or videos. Moreover, they both belong to object detection algorithms based on convolutional neural networks. Thus, they return the class and the position of the detected object. Furthermore, Detectron2 is also an instance segmentation algorithm able to return a corresponding mask that defines the area of the object.

YOLOv5 [25] comes from You Only Look Once and claims to be one of the fastest object detection algorithms. In fact, it is one of the best-known object detection algorithms due to its speed and accuracy. It is divided into three components, namely, a backbone, neck and head, as all single-stage detectors. In particular, the first component contains a backbone network that extracts rich feature representations from images. The second component consists of the model neck responsible for extracting feature pyramids that help to recognise objects of different sizes and scales. The final component is the model head that applies anchor boxes on feature maps. In addition, it is responsible for rendering the predicted class, the scores of the predicted objects, and the bounding boxes.

Detectron2 [23] is considered state-of-the-art in instance segmentation. It is the successor of Mask RCNN [30], which is built on top of Faster RCNN. Thus, apart from providing

the exact position of the detected object, it also provides a corresponding segmentation mask covering the exact area of the object.

The architecture of Mask RCNN is divided into three stages. In the first stage, a regional proposal network (RPN) is responsible for returning all regions of possible areas with detected objects. Mask RCNN uses ResNet50 or ResNet101 with FPN support as a backbone network in this stage. In the second stage, a classifier is responsible for evaluating the region proposals derived from the first stage and providing the predictions with bounding boxes for each detected object. Finally, the third stage provides the corresponding masks of the detected objects.

## 4. Materials and Methods

This section presents the methodology we followed in our research. First, we describe the image acquisition from mushrooms in the greenhouse and the annotation process. Second, we give the basic configuration of YOLOv5 for mushroom classification in three different growing stages. Moreover, we provide the architecture we used for mushroom growth monitoring.

### 4.1. Image Acquisition

Data collection took place in a greenhouse near the city of Grevena, Greece. We collected multiple images in a greenhouse, each containing one or multiple mushrooms. We captured only images that had mushrooms mainly in the foreground and not in the background because they reduced the detection accuracy. Finally, we obtained 1128 images with one or more mushrooms. We annotated them into three different classes (Stage1, Stage2, Stage3) depending on the growing stage. In more detail, the first stage corresponded to the first days of appearance, when mushrooms were too small and they had just started to form their shape. The second class corresponded to mushrooms that had already formed their shape and continued to grow. Some of them were obviously small, while others were big enough but not ready to harvest. The third class contained mushrooms that were ready to harvest, which could be manually indicated from two parameters. More specifically, when the edges of their caps started to become flat or slightly uprolled, it was an indication of their final stage of growth [31]. Figure 1 shows some examples of mushrooms belonging to the three different stages.

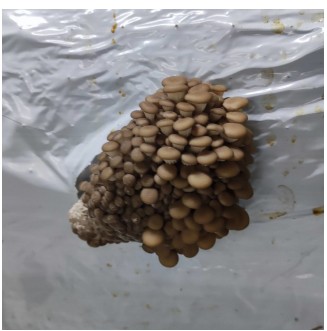 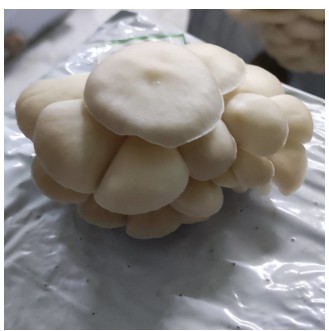 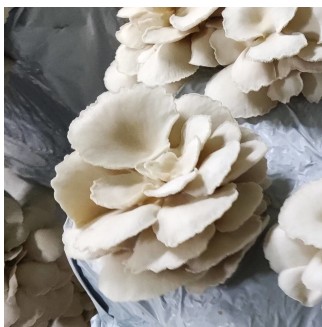

(**a**) Stage1            (**b**) Stage2            (**c**) Stage3

**Figure 1.** Examples of mushrooms in three different growth stages.

Many of the collected photos in the greenhouse contained multiple mushrooms, making the annotation complex and the recognition even more challenging, especially when the mushrooms were at the final stage and many of them were overlapping. Figure 2 presents two of those photos as an example.

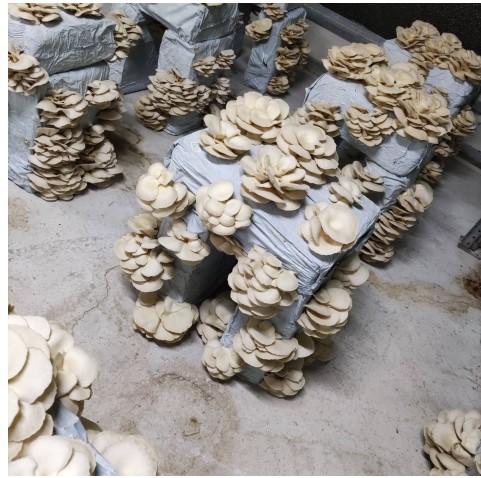
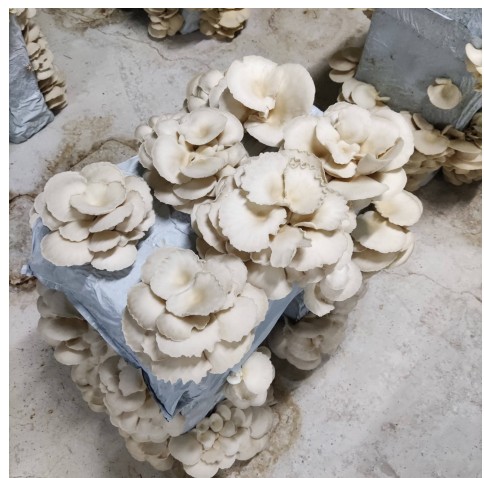

(**a**) Sample 1                          (**b**) Sample 2

**Figure 2.** Pictures from the greenhouse with multiple mushrooms.

Finally, 4271 mushrooms were annotated and classified into three different classes (Stage1, Stage2, Stage3) as described before. Moreover, 1130 of them belonged to the first growing stage, 1845 to the second growing stage, and 1296 of them to the third growing stage. In addition, the 1128 annotated photos were divided into 784 for the training dataset and 344 for the validation dataset.

For the second part of our research, we collected a different set of photos. In particular, the data acquisition for the growing rate of mushrooms was made in 33 different mushroom substrate grow bags. We captured a photo for each one on seven different days between 16 August 2022 and 24 August 2022. Finally, we collected 231 different photos in total. The distance between the camera and the substrate grow bag was one meter above. Figure 3 shows an example of three captured photos. We did not annotate these photos, but we used the trained models from Detectron2 to indicate the size of each mushroom. In our evaluation for mushroom growth monitoring, we used only the mushrooms that were on top of the substrate grow bag.

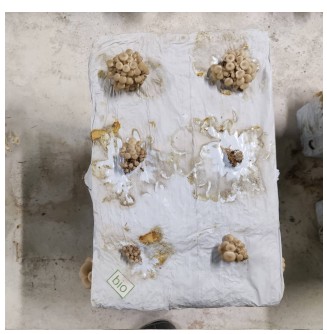
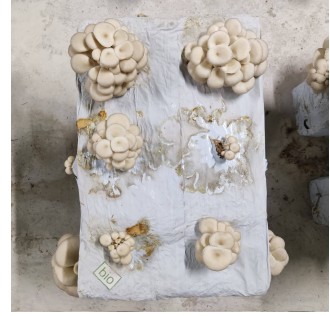
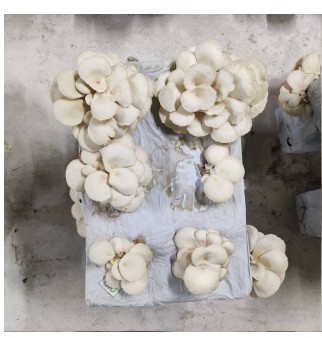

(**a**) Captured on 18/8/2022     (**b**) Captured on 19/8/2022     (**c**) Captured on 22/8/2022

**Figure 3.** Examples of mushroom substrate grow bag in three different days.

For training Detectron2 for instance segmentation, we annotated another two datasets. The first one contained annotations for substrate grow bags, and the second one contained annotations for mushrooms. The annotation process for the image segmentation is a time-consuming task since we had to annotate precisely the objects with polygons. For that reason, we chose to annotate only photos that contained a small number of mushrooms to simplify the annotation. Finally, we annotated 453 mushrooms and divided them into 358 for the training dataset and 95 for the validation dataset. In addition, for the detection of substrate grow bags, we annotated 200 photos and divided them into 150 for the training dataset and 50 for the validation dataset.

### 4.2. Configuration and Method of Mushroom Growth Stage Detection

For mushroom detection and classification in the three different classes as described previously, we used YOLOv5. First, we trained our models with the default hyperparameters using the pretrained models YOLOv5s and YOLOv5l with an image size of 640 pixels and batch sizes of 2, 4, and 8.

To achieve better results, we used the provided method integrated with YOLOv5 for hyperparameter optimisation called hyperparameter evolution. It is a genetic algorithm to find the best set of hyperparameters for the specific dataset. The evolving procedure starts from the default hyperparameters or other user-defined values, if available, and tries to improve a fitness function in every loop. The default fitness function is a weighted combination of mAP@0.5 with 10% contribution and mAP@0.5:0.95 with 90% contribution. In every evolving loop, the primary genetic operator is the mutation. The proposed combination for the mutation uses 80% probability and a 0.04 variance to calculate the next combination of hyperparameters based on the best parents from previous generations.

We used the hyperparameter evolution approach for the two pretrained models, YOLOv5s and YOLOv5l, and trained them for 600 generations with an image size of 640 pixels and a batch size of 4. Table 1 shows the predicted sets of hyperparameters for each pretrained model.

**Table 1.** Optimised hyperparameters after evolving for YOLOv5s and YOLOv5l.

| Hyperparameter | YOLOv5s | YOLOv5l |
|---|---|---|
| lr0 | 0.01193 | 0.00635 |
| lrf | 0.01110 | 0.01358 |
| momentum | 0.94993 | 0.94506 |
| weight_decay | 0.00058 | 0.00058 |
| warmup_epochs | 2.9186 | 4.4758 |
| warmup_momentum | 0.88378 | 0.95 |
| warmup_bias_lr | 0.08423 | 0.05706 |
| box | 0.06250 | 0.04464 |
| cls | 0.66883 | 0.64453 |
| cls_pw | 1.1594 | 0.95558 |
| obj | 1.1757 | 1.1124 |
| obj_pw | 0.94963 | 0.96568 |
| iou_t | 0.2 | 0.2 |
| anchor_t | 4.1487 | 3.4772 |
| fl_gamma | 0.0 | 0.0 |
| hsv_h | 0.01217 | 0.01939 |
| hsv_s | 0.85528 | 0.57067 |
| hsv_v | 0.37136 | 0.38627 |
| degrees | 0.0 | 0.0 |
| translate | 0.10421 | 0.09902 |
| scale | 0.47879 | 0.43253 |
| shear | 0.0 | 0.0 |
| perspective | 0.0 | 0.0 |
| flipud | 0.0 | 0.0 |
| fliplr | 0.5 | 0.5 |
| mosaic | 0.98804 | 0.92192 |
| mixup | 0.0 | 0.0 |
| copy_paste | 0.0 | 0.0 |
| anchors | 2.0763 | 2.6089 |

Figure 4 shows a graphical representation for YOLOv5l with each hyperparameter displayed in a different subplot. Each subplot presents all values from all generations for the specific hyperparameter. The horizontal axis corresponds to the value of the hyperparameter, and the vertical value corresponds to the calculated fitness. The yellow areas indicate a high concentration of values. Subplots with vertical distributions indicate that the specific hyperparameter was disabled and did not mutate.

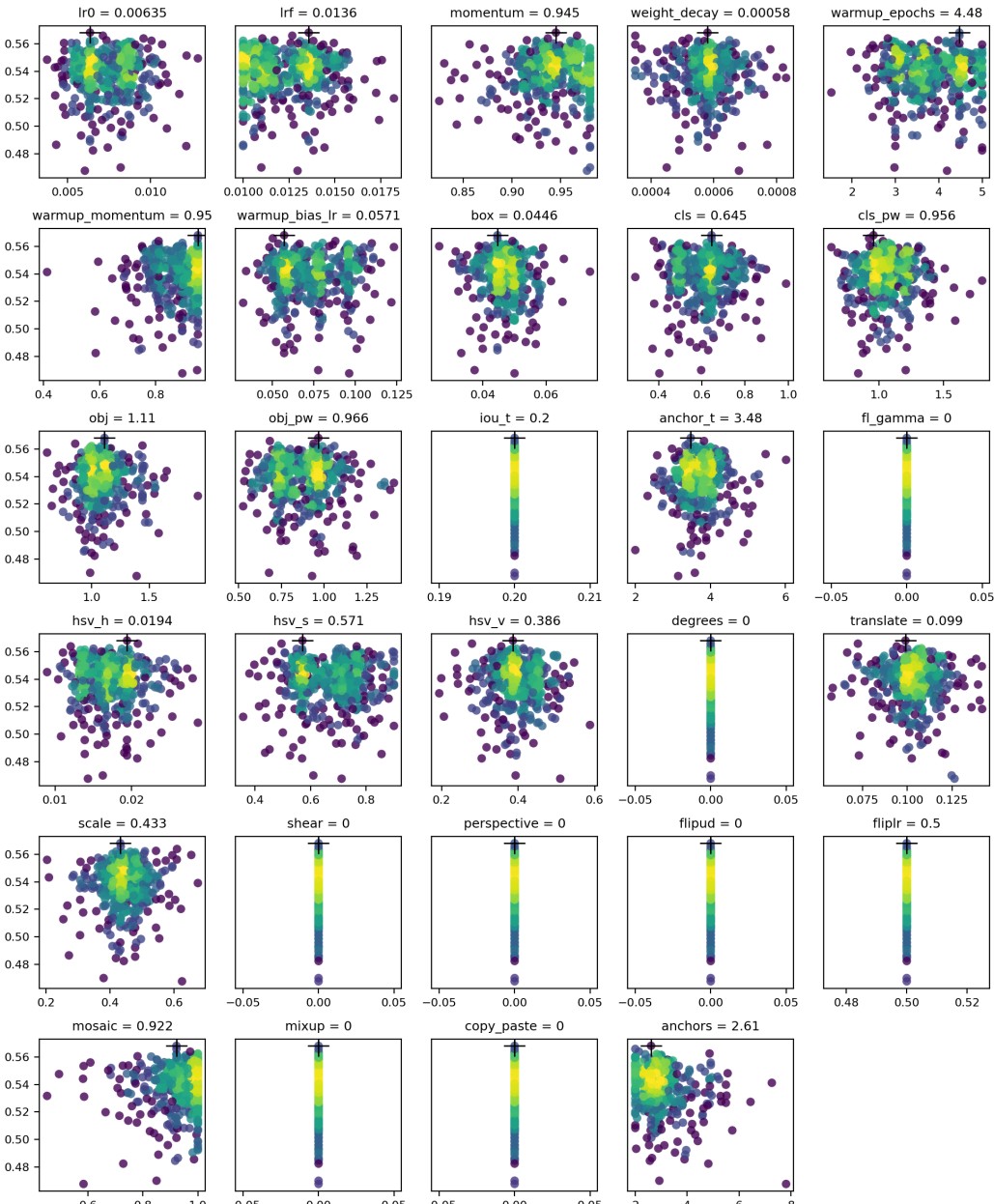

**Figure 4.** Graphical representation for the calculated values of all hyperparameters, after evolving for 600 generations for YOLOv5l.

To evaluate the results of different configurations of YOLOv5, we used the metrics mAP@0.5, mAP@0.5:0.95, precision, recall, and F1-score.

The mAP@0.5 metric corresponds to the average precision over a threshold of 0.5 for the object detection. In addition, the mAP@0.5:0.95 denotes the average precision with a threshold between 0.5 and 0.95.

The precision metric indicates the correctly identified trees divided by the total number of detected trees and is given by Equation (1). The recall metric indicates the falsely identified trees divided by the total number of actual existing trees and is given by Equation (2).

$$Precision = \frac{TP}{FP + TP} \tag{1}$$

$$Recall = \frac{TP}{FN + TP} \tag{2}$$

where $TP$ (true positive) is the number of correctly detected objects in the class, $FP$ (false positive) is the number of falsely detected objects in the specific class, and $FN$ (false negative) is the number of objects that are not detected in the specific class.

The metric *F1-score* is widely used as one of the main metrics for the accuracy of object detection algorithms, and it is given by Equation (3).

$$F1\ Score = 2 \times \frac{P \times R}{P + R} \qquad (3)$$

### 4.3. Configuration and Method of Mushroom Growth Monitoring

For mushroom growth monitoring, we used two different stages for the object detection. The first one was responsible for detecting the substrate grow bag, and the second one was responsible for detecting the mushrooms in a photo. Figure 5 illustrates the main components of the architecture of the proposed method.

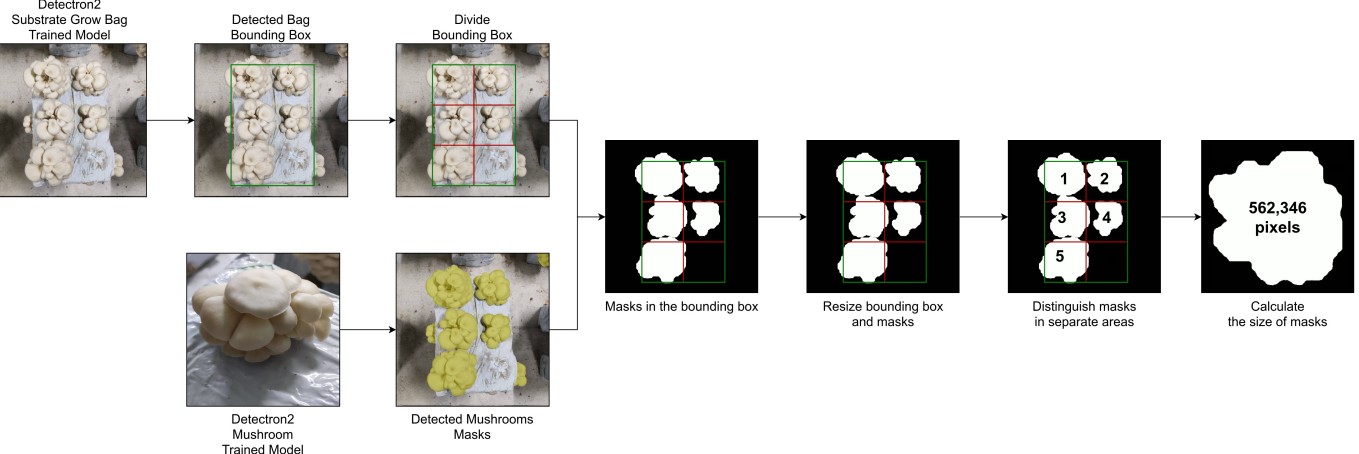

**Figure 5.** Main components of the architecture for mushroom growth monitoring.

More specifically, in the stage of the substrate grow bag detection, we use the trained model with Detectron2. We only obtained the bounding box of the detected substrate grow bag and not the mask. After the detection of the substrate grow bag, we used the corresponding bounding box and divided it into six equal rectangles (2 × 3). Figure 6a illustrates the bounding box (green) and the lines (red) that divide the bounding box in order to distinguish the detected mushrooms.

Moreover, we used the trained model with Detectron2 to detect the mushrooms in the photo and return the corresponding masks. Figure 6b displays the detected mushrooms. Next, we performed a resize operation on the bounding box and the masks to achieve normalisation for calculating the size in all photos of the same substrate grow bag. Furthermore, the detected mushrooms were distinguished in one of the rectangles with the following rules. A mushroom should have more than 50% in the specific rectangle. Only one mushroom can belong in one rectangle, and each mushroom belongs only in one rectangle. In any case, the selected mushroom was the one that covered the most area in the rectangle.

Finally, Figure 6c illustrates the divided bounding box with the corresponding masks of all detected masks. The size of the provided mask could indicate the size of the mushroom. After distinguishing each mushroom, we calculated the coverage based on the pixels of the provided mask and used it for mushroom growth monitoring.

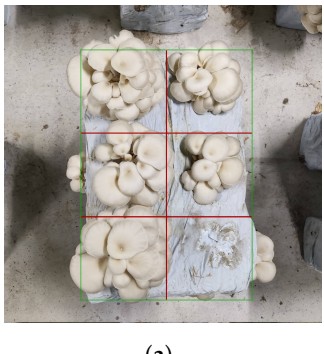 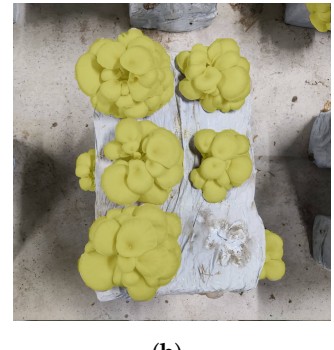 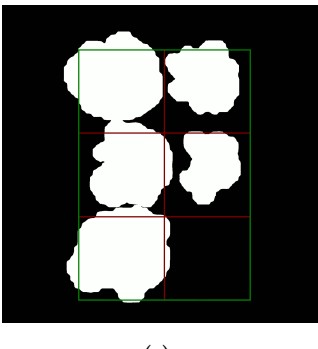

(**a**)          (**b**)          (**c**)

**Figure 6.** Example photos of mushroom growth monitoring. (**a**) Bounding box divided into six equal rectangles; (**b**) Masks of the detected mushrooms; (**c**) Masks in the six rectangles of the bounding box.

## 5. Results

In the following subsections, we evaluate the results of the mushroom classification for different growth stages. Next, we provide the results of mushroom monitoring for their growth rate.

### 5.1. Evaluation of Mushroom Growth Stage Detection

In this subsection, we compared different configurations of YOLOv5 for mushroom detection and classification in three classes, as described previously. We used the metrics mAP@0.5, mAP@0.5:0.95, precision, recall, and F1-score to compare the accuracy of each trained model.

Table 2 shows a comparison of all configurations. More specifically, we use the pretrained models YOLOv5s and YOLOv5l with the default (D) configuration for the hyperparameters and with the configuration derived from evolution (E). Moreover, our configurations used an image size of 640 pixels and three different values for the batch size (2, 4, 8).

**Table 2.** Evaluation results for mAP@0.5 and mAP@0.5:0.95 for different hyperparameter configurations (HC), for default values (D) and optimised values with evolution (E).

| Pre-Trained Model | HC | Image Size | Batch Size | mAP@0.5 | mAP@0.5:0.95 |
|---|---|---|---|---|---|
| YOLOv5s | D | 640 | 2 | 0.79230 | 0.53119 |
| YOLOv5s | E | 640 | 2 | 0.79532 | 0.53405 |
| YOLOv5s | D | 640 | 4 | 0.78946 | 0.53071 |
| YOLOv5s | E | 640 | 4 | 0.79307 | 0.53541 |
| YOLOv5s | D | 640 | 8 | 0.78006 | 0.52299 |
| YOLOv5s | E | 640 | 8 | 0.78457 | 0.52511 |
| YOLOv5l | D | 640 | 2 | 0.77675 | 0.52999 |
| YOLOv5l | E | 640 | 2 | 0.79072 | 0.53476 |
| YOLOv5l | D | 640 | 4 | 0.78027 | 0.53390 |
| YOLOv5l | E | 640 | 4 | 0.78261 | 0.53810 |
| YOLOv5l | D | 640 | 8 | 0.79166 | 0.54008 |
| YOLOv5l | E | 640 | 8 | 0.79353 | 0.54582 |

It seems that there was not much difference in all configurations. Based on mAP@50 and mAP@50:95, configurations with hyperparameters derived from evolution showed a slightly better performance. Moreover, configurations based on YOLOv5l returned a slightly better performance. The model based on YOLOv5l with hyperparameters derived from evolution and with a batch size of eight achieved the best performance. More specifically, it reached an mAP@0.5 of 0.79353 and an mAP@0.5:0.95 of 0.54582.

Another type of evaluation was based on the F1-Score metric. Table 3 shows the comparison of all configurations based on the best occurrence of the F1-Score. Better performance on this metric was achieved by the same configuration with YOLOv5l, hyperparameters derived from evolution, and a batch size of eight, as it reached an F1-Score of 76.65%.

**Table 3.** Evaluation results for the F1-Score for different hyperparameter configurations (HC), for default values (D) and optimised values with evolution (E).

| Pre-Trained Model | HC | Image Size | Batch Size | Precision | Recall | F1-Score |
|---|---|---|---|---|---|---|
| YOLOv5s | D | 640 | 2 | 76.54% | 74.50% | 75.51% |
| YOLOv5s | E | 640 | 2 | 74.68% | 76.51% | 75.59% |
| YOLOv5s | D | 640 | 4 | 74.82% | 75.95% | 75.38% |
| YOLOv5s | E | 640 | 4 | 76.55% | 74.50% | 75.51% |
| YOLOv5s | D | 640 | 8 | 76.54% | 73.58% | 75.03% |
| YOLOv5s | E | 640 | 8 | 74.94% | 75.51% | 75.22% |
| YOLOv5l | D | 640 | 2 | 74.95% | 75.94% | 75.44% |
| YOLOv5l | E | 640 | 2 | 75.56% | 75.66% | 75.61% |
| YOLOv5l | D | 640 | 4 | 77.30% | 73.28% | 75.24% |
| YOLOv5l | E | 640 | 4 | 73.60% | 77.22% | 75.37% |
| YOLOv5l | D | 640 | 8 | 75.45% | 75.73% | 75.59% |
| YOLOv5l | E | 640 | 8 | 77.89% | 75.44% | 76.65% |

Figure 7 shows the curve of the F1-Score for the best configuration. The horizontal axis corresponds to the confidence of the detected mushrooms, while the vertical axis corresponds to the F1-score. In this example, the maximum F1-score had a confidence of 0.566. In addition, the graph shows that the class "Stage1" had a better accuracy. This was expected since all photos with mushrooms belonging to "Stage1" were simpler, as mushrooms were small and not overlapping.

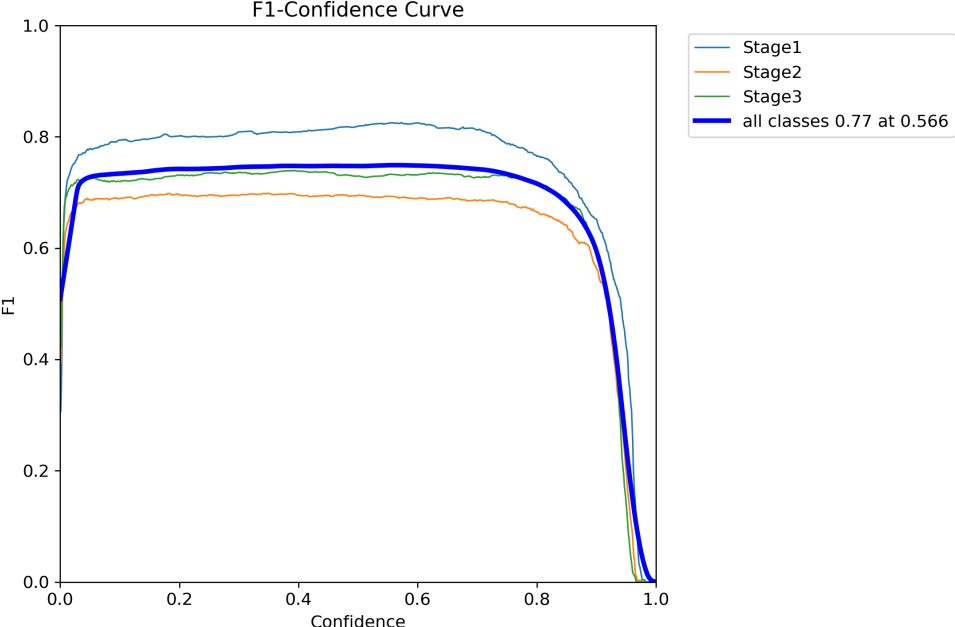

**Figure 7.** F1 curve.

To notice the detection accuracy for each class more clearly, Figure 8 presents the confusion matrix for the same configuration. It shows that class "Stage1" had an accuracy

of 82%, class "Stage2" had an accuracy of 71%, and "Stage3" had an accuracy of 70%. Although this does not seem perfect, we can say that it was good enough depending on the complexity of the photos with mushrooms in "Stage2" and "Stage3".

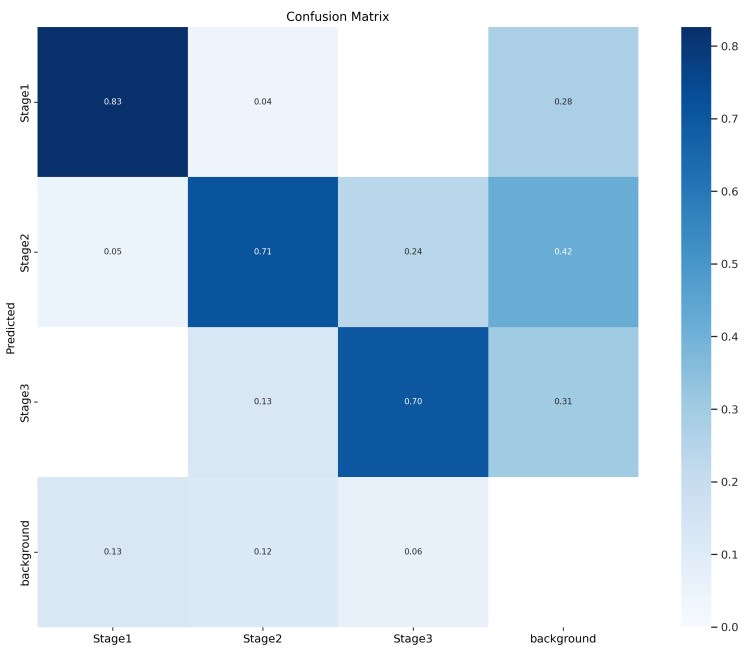

**Figure 8.** Confusion matrix for all mushroom growing stages.

Overall, the evaluation results were acceptable for detecting different stages of mushrooms.

Figure 9 shows an example of the detection where only mushrooms from classes Stage2 and Stage3 exist. The green bounding box indicates that the mushroom is ready to harvest (Stage3). The blue bounding box indicates that the mushroom is not yet ready to harvest (Stage2). We marked in magenta the mushrooms that were detected in the wrong class. Moreover, we marked in red the mushrooms that were not detected. It seems from the photo that the mushrooms in the foreground were detected in the right class, while some errors occurred in mushrooms that were in the background or partially displayed.

### 5.2. Evaluation of Mushroom Growth Monitoring

After the procedure described previously, we obtained 575 different masks from 114 different mushrooms in the greenhouse. All masks were distinguished for different mushrooms depending on the number of the substrate grow bag and the position on it. When the mask corresponded to the first day of appearance of each mushroom, it was marked as day one. The corresponding masks of the next days of the same mushroom were numbered as well in the same way.

Figure 10 shows the growth rate of four different mushrooms. Each mushroom is indicated with a different colour. The horizontal axis of the graph contains the number of the growth day, starting from number one for the first day of the mushroom's appearance. The vertical axis indicates the size of the mushroom counted in pixels.

Moreover, we calculated the maximum size of each mushroom based on all photos before it was harvested. We set its size as 100%, and we calculated the size of the same mushroom for all other days accordingly. Figure 11 shows the growth rate of the same mushrooms but as a percentage of the maximum size of each mushroom.

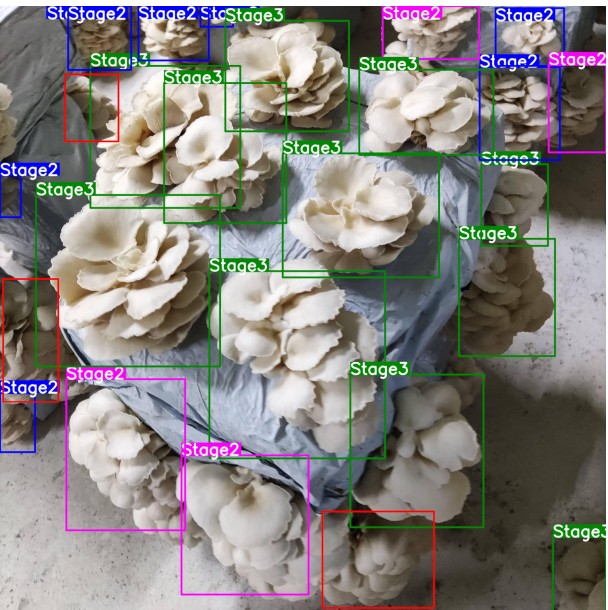

**Figure 9.** Example of mushroom detection in three different stages.

**Figure 10.** Comparing the absolute size of the growth rate of mushrooms.

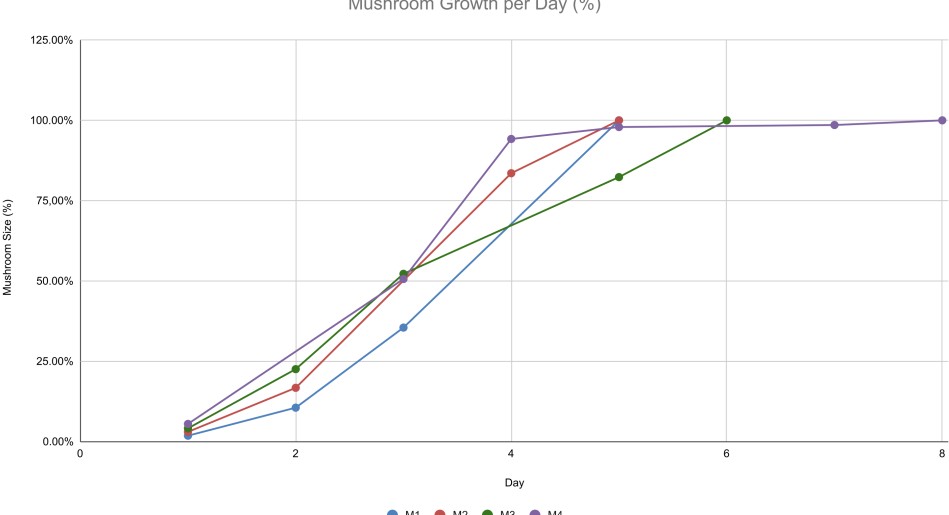

**Figure 11.** Comparing the percentage of the growth rate of mushrooms.

Both graphs show that the three mushrooms (M1, M2, M3) followed an almost linear growth rate. The fourth mushroom (M4) also followed a linear growth rate for the first four days, but from the fifth day, it seemed not to change in size. In fact, on the fifth day, that specific mushroom should have been harvested, but it was accidentally forgotten. Figure 12 shows the captured photos of this mushroom (M4), from which it seemed that the mushroom was ready to harvest on day five.

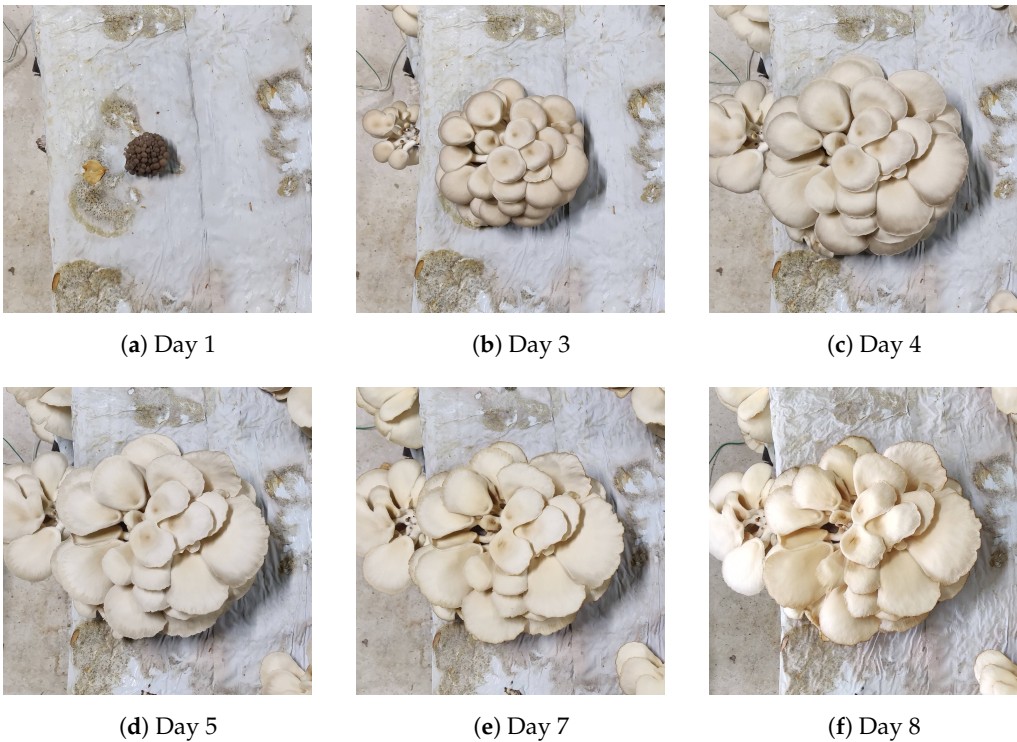

|  |  |  |
|:---:|:---:|:---:|
| (**a**) Day 1 | (**b**) Day 3 | (**c**) Day 4 |
| (**d**) Day 5 | (**e**) Day 7 | (**f**) Day 8 |

**Figure 12.** Mushroom growth example photos.

We manually evaluated the harvesting status of each mushroom and decided that 17 mushrooms could be harvested earlier, which represented 14.04% of the mushrooms. The average difference between the size of the masks from the day identified as ready to harvest and the previous day was 5.34%. Thus, if we set it as a threshold in our method to provide an alert any time that the difference in the mushroom size was below it, we could inform the farmer of the harvest time. This would lead to an improvement in the quality of the harvested mushrooms as we would avoid overripe mushrooms. In addition, our evaluation showed that the average days for the growth of mushrooms was 5.22.

Figure 13 displays the graph with all mushrooms detected for all days. All values are displayed in percentage of the maximum size of the specific mushroom. The blue marks indicate the detected mushrooms that were not ready to harvest, while the red marks indicate mushrooms that should have been harvested at least one day before.

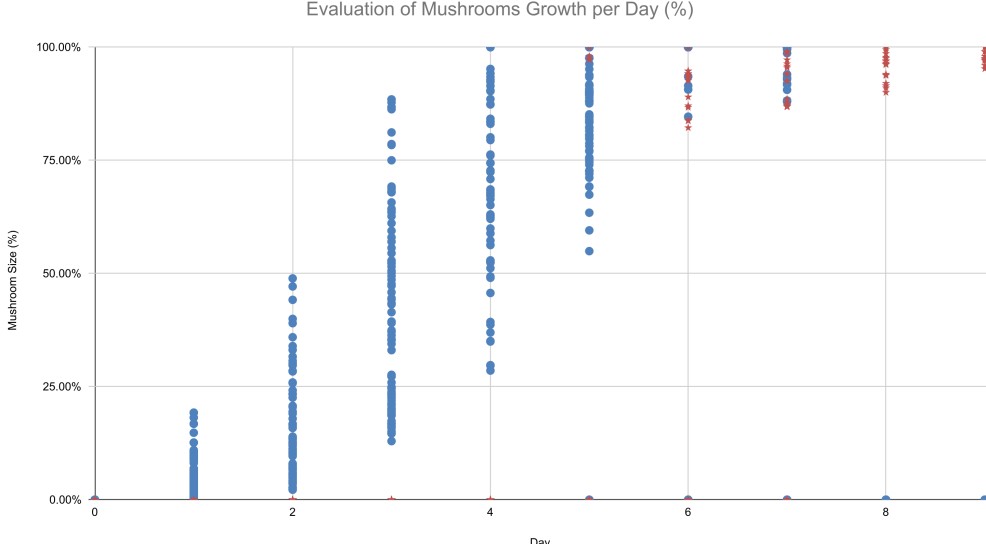

**Figure 13.** Percentage of the growth rate of all mushrooms. Red marks indicate mushrooms that should have been harvested earlier.

## 6. Discussion

Our first approach for mushroom detection with YOLOv5 could be a valuable part of a decision support system. For example, it may be useful in tasks such as yield prediction or product improvement by indicating those mushrooms that are ready for harvest. Overall, the accuracy of the trained model was considered acceptable regarding the complexity of the photos of the dataset.

However, our experience shows that some changes could improve the results. A more precisely selected dataset should improve the detection accuracy of different classes. For example, photos with fewer mushrooms and only in the foreground will give more accurate results. As we discussed in Figure 9, the main drawback in detection accuracy is that many mushrooms are partially visible in the photos, and many of them are in the background, which makes it difficult to detect and classify because their features are not clear. Moreover, a proposed methodology to overcome these obstacles could be to exclude mushrooms that are partially displayed in the picture and to focus only on those clearly displayed.

Furthermore, the 600 steps we used for the hyperparameter evolution in this research may not be good enough, and more steps will result in a better detection accuracy. This would lead to better combinations of the hyperparameters, but the hyperparameter evolution procedure is a very time-consuming task, even for a professional graphic processing unit. A better approach to enhance the model's accuracy is to modify the YOLOv5 algorithm. Thus, the trained model would include the specific characteristics of the different classes.

Our second approach for mushroom growth monitoring could also be helpful for a decision support system to inform farmers about the harvesting time. As a result, an improvement in product quality is expected since mushrooms will be collected at the right time. Furthermore, the precisely detected mask of the mushrooms may support other potential applications. For example, it could be a useful part of a robotic mechanism in order to detect, decide, and collect only those mushrooms that are ready to harvest without damaging them.

The method of mushroom growth monitoring with Detectron2 also has some limitations. Our method focused only on mushrooms that were on top of the substrate bag, which was placed horizontally in the greenhouse. In fact, only one-third of the mushrooms could be observed with this method since there were also substrate bags positioned vertically below the substrate bag we used. In addition, in many cases, the substrate bags are densely placed in the greenhouse, and sometimes, the farmers use more than one layer of substrate grow bags in the greenhouse, making it difficult to capture photos even from those positioned horizontally.

Furthermore, getting only one image per day has the disadvantage that the growing days may not be counted precisely in some mushrooms. For example, some mushrooms may appear a few hours after a picture was taken, resulting in a day count less than the actual growth duration. Moreover, some mushrooms may have been harvested just before a picture was taken, also resulting in reduced count days. Therefore, capturing photos of the substrate bags more often could give better results and inform more precisely about the growing rate of the mushroom and the harvesting time.

Finally, the results of the mushroom detection are not always precise. In fact, the proposed method has two minor drawbacks. First, some of the mushrooms that were too small could not be detected. In addition, the masks returned from Detectron2 were not always accurate. We believe that both these drawbacks do not have a large impact on the proposed methodology. However, if we need to improve them, we could use a larger dataset with more annotated mushrooms to get better results.

## 7. Conclusions

Object detection and instance segmentation have multiple applications in various domains. In smart farming, it is mainly used in fruit detection, weed detection, or pest detection.

In this paper, we evaluated the effectiveness of YOLOv5 and Detectron2 in mushroom detection in a greenhouse. Firstly, we evaluated YOLOv5 on its effectiveness in classifying mushrooms in three growth stages. The results showed that even in complex environments such as a greenhouse with *Pleurotus* mushrooms, it was possible to identify mushrooms that were ready to harvest. The evaluation results on mushroom detection and classification in three different growing stages gave an F1-score of up to 76.5%, and specifically for the final growing stage, an accuracy of up to 70%.

Secondly, we proposed a method for mushroom growth monitoring. For that purpose, we used two trained models with Detectron2. The first one detected the substrate grow bag, and the second detected the mushrooms and returned the corresponding masks. Experimental results showed that the growth rate of *Pleurotus* mushrooms was linear. Moreover, this procedure made it possible to detect when mushrooms reached their maximum size and were ready to harvest. The evaluation results showed that it was possible to make decisions and improve harvesting time for up to 14.04% of the mushrooms in the greenhouse. In addition, the results showed that, on average, *Pleurotus* mushrooms needed 5.22 days to reach maximum size.

Moreover, the proposed methods are suitable to be part of a decision support system to inform farmers about the status of their cultivation. In addition, a future potential application would be a robotic mechanism able to detect and recognise the growth stage of mushrooms before harvesting them.

**Author Contributions:** Conceptualization, V.M. and P.S.; methodology, V.M. and G.K.; software, V.M.; validation, G.K. and P.S.; formal analysis, V.M.; investigation, V.M.; resources, V.M. and N.M.; data curation, V.M. and G.K.; writing—original draft preparation, V.M.; writing—review and editing, S.B. and I.M.; visualization, V.M.; supervision, P.S.; project administration, P.S.; funding acquisition, P.S. All authors have read and agreed to the published version of the manuscript.

**Funding:** This research was cofunded by the European Regional Development Fund of the European Union and Greek national funds through the Operational Program Western Macedonia 2014–2020, under the call "Collaborative and networking actions between research institutions, educational institutions and companies in priority areas of the strategic smart specialization plan of the region", project "Smart Mushroom fARming with internet of Things—SMART", project code: DMR-0016521.

**Institutional Review Board Statement:** Not applicable.

**Data Availability Statement:** The data presented in this study are available on request from the corresponding author. The data are not publicly available due to restrictions on privacy.

**Conflicts of Interest:** The authors declare no conflict of interest.

## Abbreviations

The following abbreviations are used in this manuscript:

| | |
|---|---|
| YOLO | You Only Look Once |
| RCNN | Region-based convolutional neural network |
| RPN | Regional proposal network |
| FPN | Feature pyramid network |
| mAP | mean average precision |
| TP | True positive |
| FP | False positive |
| FN | False negative |
| DT | Decision tree |
| RGB | Red green blue |
| LR | Logistic regression |
| KNN | K-nearest neighbours |
| SVM | Support vector machine |
| NB | Naive Bayes |
| RF | Random forest |

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
