# Peer review of "Monitoring Mushroom Growth with Machine Learning"

_agriculture, doi:10.3390/agriculture13010223_

Round 1

Reviewer 1 Report

·      The paper includes a clear and detailed explanation of the methods used to conduct the research, including any experimental setup or data collection procedures.

·      The paper clearly explains any technical terms or jargon that may be unfamiliar to readers outside the field.

·      The paper includes a thorough review of relevant literature and situates the work within the broader context of the field.

·      I really enjoyed reading your manuscript, evaluating the effectiveness of object detection and instance segmentation in Smart Farming Intervention, and I think it is a valuable contribution to the field. The methods you used to design and evaluate your intervention are well thought-out and the results are clearly presented. However, as I read through your manuscript, I noticed that you did not discuss the potential limitations of your study. While I understand that you may not have been able to control for all variables that could impact the results of your intervention, it is important to acknowledge these limitations in order to provide a completer and more accurate picture of your work.

I suggest that you consider adding a section to your manuscript that discusses the limitations of your study and how they may have impacted your results. This could include things like the small sample size, the limited generalizability of your findings, or the potential for bias in your data collection or analysis.

Overall, I believe that your work is of high quality and that addressing the limitations of your study will only strengthen your contributions to the field.

Reviewer 2 Report

The manuscript describes how machine learning can be used for mushroom growth monitoring, where YOLOv5 is used to detect the growth stage of the mushroom and Detectron2 to monitor the growth of the mushroom. The idea is relatively new, but there are some minor issues with the article. My detailed comments are as follows:

Comments 1: The use of words in the manuscript is rather homogeneous, e.g., present, more specifically. In the introduction part, the logical relationship between sentences needs to be think over, it is recommended that the authors rewrite the introduction and emphasize the importance of this study.

Comment 2: There are many details in the manuscript that need to be corrected, such as spelling, grammar, upper- and lower-case letters, references, etc.

Comment 3: The number of pictures in the manuscript is too much, and the author is advised to put the not very important pictures in the supplementary material.

Comment 4: Lines 233-247, the evaluation indicators for the model could be in the “Materials and Methods” section for the reader's understanding.

Comment 5: Suggest adding more discussion and outlook to the manuscript, explore whether there are differences with the findings of others, and explore the limitations and breakthroughs in your study.

Round 2

Reviewer 2 Report

Comment 1. Figure 7.F1 curve and line 288 should have a space. Similar problems exist in other places.